# Nitrogen-Doped Pitch-Based Activated Carbon Fibers with Multi-Dimensional Metal Nanoparticle Distribution for the Effective Removal of NO

**Shengkai Chang [1], Zhuo Han [1], Jianxiao Yang [1],\*, Xuli Chen [1], Jiahao Liu [1], Yue Liu [1] and Jun Li [2],\***

[1] Hunan Province Key Laboratory for Advanced Carbon Materials and Applied Technology, College of Materials Science and Engineering, Hunan University, Changsha 410082, China
[2] School of Chemistry and Biological Engineering, Changsha University of Science and Technology, Changsha 410114, China
\* Correspondence: yangjianxiao@hnu.edu.cn (J.Y.); muzi.yikou@csust.edu.cn (J.L.); Tel./Fax: +86-731-8882-2733 (J.Y.)

**Abstract:** The design of catalytic materials for $NO_X$ removal by the selective catalytic reduction with $NH_3$ ($NH_3$-SCR) has been a focus of research in the field of waste gas treatment. In this work, pitch-based activated carbon fibers (ACFs) were impregnated with copper nitrate and cerium nitrate, and then the ACFs that were loaded with bimetallic nanoparticles (ACF@Cu/Ce) were obtained after the pyrolyzation and reduction were performed. Moreover, the ACF@Cu/Ce were furtherly treated through the chemical vapor deposition and $NH_3$ activation, through which the N-doped carbon nanofibers (N-CNFs) were grown on the surface of the ACFs. Thus, the catalytic material with a multi-dimensional metal nanoparticle distribution and nitrogen-rich network structure, namely the N-CNF/ACF@Cu/Ce, was constructed. In the $NH_3$-SCR reaction, the NO conversion of the N-CNF/ACF@Cu/Ce could be maintained at about 72~81% in a wide temperature window of 295~495 °C, which enabled the N-CNF/ACF@Cu/Ce to meet the requirements of the practical applications.

**Keywords:** activated carbon fibers; $NH_3$-SCR catalysts; multi-scale distribution; metal nanoparticles; $NO_X$





## 1. Introduction

$NO_X$ from automobile exhausts and industrial emissions have become one of the most serious environmental problems. Among all of the methods to remove $NO_X$, the selective catalytic reduction of it with $NH_3$ ($NH_3$-SCR) has the great advantages of a low cost and a high efficiency, therefore, it is considered as an ideal way to remove $NO_X$, and it is widely applied for $NO_X$ removal in the range of 300~400 °C [1–5]. Generally, there are two reaction mechanisms of $NH_3$-SCR, namely, the E–R mechanism and the L–H mechanism [3,6]. The former refers to the formation of active intermediate species between the adsorbed $NH_3$ and gas NO, plus the subsequent decomposition of them to form $N_2$ and $H_2O$, while the latter refers to the reaction between the adsorbed $NH_3$ and adsorbed NO [3,7,8]. At present, $V_2O_5/WO_3(MoO_3)$-$TiO_2$ catalysts for the SCR have already been used commercially in thermal power plants and other facilities, which can already achieve the NO conversion that is higher than 90% [9,10]. However, their application is quite limited due to the narrow temperature window of 300~400 °C that is used, their low thermal stability and their high biological toxicity [11,12]. Moreover, the vanadium-based catalysts can be easily polluted by $SO_2$, dust or volatiles, which leads to the deactivation of the catalyst [10], and so researchers have been working on developing alternative vanadium-free SCR catalysts.

The $CeO_2$ catalyst has been studied extensively in recent years in catalytic oxidation, fuel cells and other fields. It is indicated that Ce has two kinds of valence states (+3/+4), which not only makes $CeO_2$ have a large oxygen storage capacity and an excellent redox

capacity [3,13], but also affects the surface reconstruction and the electron transfer between the reactant molecules and the catalyst. Actually, the $NH_3$-SCR activity of pure $CeO_2$ is poor. However, it is reported that by combining $CeO_2$ with certain metal nanoparticles, like copper, the catalysts can reach much higher SCR activity values than they could by using one of them alone, thereby indicating the synergistic effect of the nanocomposite catalysts in the SCR [14–16].

Further, the whole process of $NO_X$ purification from the atmosphere includes not only NO conversion but also the former step of $NO_X$ capture and adsorption. It should be noted that activated carbon fibers (ACFs) have several critical advantages such as high adsorption–desorption rates, a high adsorption capacity, excellent physical and chemical properties, a large specific surface area and rich surface functional groups, and ACFs can endure quite high temperatures in an anoxic environment, thus making ACFs a perfect choice for flue gas adsorption and catalyst carriers. Gupta et al. [17] prepared a catalyst with a three-dimensional network of Cu-doped ACFs and carbon nanofibers, thereby showing the high activity of $NO_X$ removal. Moreover, surface modification processes, such as surface oxidation, alkali activation and $NH_3$ activation, can not only further improve the adsorption of the pollutant ions and toxic gases by changing the surface morphology of the ACFs, but also introduce the functional groups on the surface of the ACFs to change their physical and chemical properties, thus increasing the chemical bonding between the interfaces and improving the polarity and catalytic activity [18–20].

In this work, an efficient catalytic material with a multi-dimensional metal nanoparticle distribution and a nitrogen-rich network structure was constructed, which had a large NO adsorption capability, a high reactivity and a wide operable temperature window of $NH_3$-SCR, and a high thermal stability in the flue gas atmosphere, thus contributing to the design of catalytic materials for a wide-temperature SCR.

## 2. Results and Discussion

The pore structure of the fiber surface and the dispersion of the metal nanoparticles were observed by scanning electron microscope (SEM), and when it was combined with the Energy Dispersive Spectroscopy (EDS) in situ analysis, the compositions of the nanoparticles were analyzed. As shown in Figure 1b–d, the metal particles were successfully loaded on the CNF/ACF surface. Cu and Ce, the Lewis acids, could coordinate with the oxygen atoms or the nitrogen atoms of the functional groups [16,21] on the ACF surface that was modified by $HNO_3$, thus it was well-loaded. The particles were more evenly distributed and had a smaller and more uniform size by adding the Cu and Ce together. Meanwhile, CNFs were successfully grown on the ACFs by adding Cu or Cu/Ce, thereby forming a 3D network structure. The CNFs were thought to be grown by the tip-growth mechanism [13], i.e., the Cu nanoparticles were pushed out of the ACF surface due to the nucleation and growth of the CNFs, and they were kept wrapped on the CNF tips as the CNFs grew, thus playing a key catalytic role while the $CeO_{2-x}$ nanoparticles stayed distributed in the ACFs.

The EDS analysis results of the corresponding area of the images that were taken by the SEM method are shown in Figure 1(b-1,c-1,d-1). The atomic percentages of Ce or Cu in the N-CNF/ACF@Ce and N-CNF/ACF@Cu were 1.77% and 0.95%, respectively, and the atomic percentages of Cu and Ce in the N-CNF/ACF@Cu/Ce were 0.33% and 0.15%, respectively, which proved the composition of the corresponding metal nanoparticles. Therefore, the catalytic material N-CNF/ACF@Cu/Ce successfully realized the characteristics of the multi-dimensional metal nanoparticle distribution and a satisfactory 3D network structure of the fibers.

The thermogravimetry (TG) and differential scanning calorimetry (DSC) were conducted in the air to evaluate the approximate content of the metal nanoparticles that were loaded on the fiber surface and to analyze the characteristics of the oxidation process. As shown in Figure 2a, the TG curve of the N-CNF/ACF@Ce showed that there was a weight increase at the beginning of the heating. This might be because Ce could easily coordinate to oxygen, which promoted oxygen to get into the fibers. The N-CNF/ACF@Ce had the

largest residual quantity, indicating that is had the largest content of metal nanoparticles. However, the residual quantity of the N-CNF/ACF@Cu/Ce was slightly larger than that of the N-CNF/ACF@Cu, which could also be attributed to the increase in oxygen. The corresponding DSC curves are given in Figure 2b. The N-CNF/ACF@Cu/Ce had the lowest reaction peak temperature of 385 °C, which was followed by those of N-CNF/ACF@Ce and then, N-CNF/ACF@Cu, indicating that it had the highest reactivity with oxygen, which confirmed the oxygen affinity of Ce and the synergistic effect of Cu/Ce.

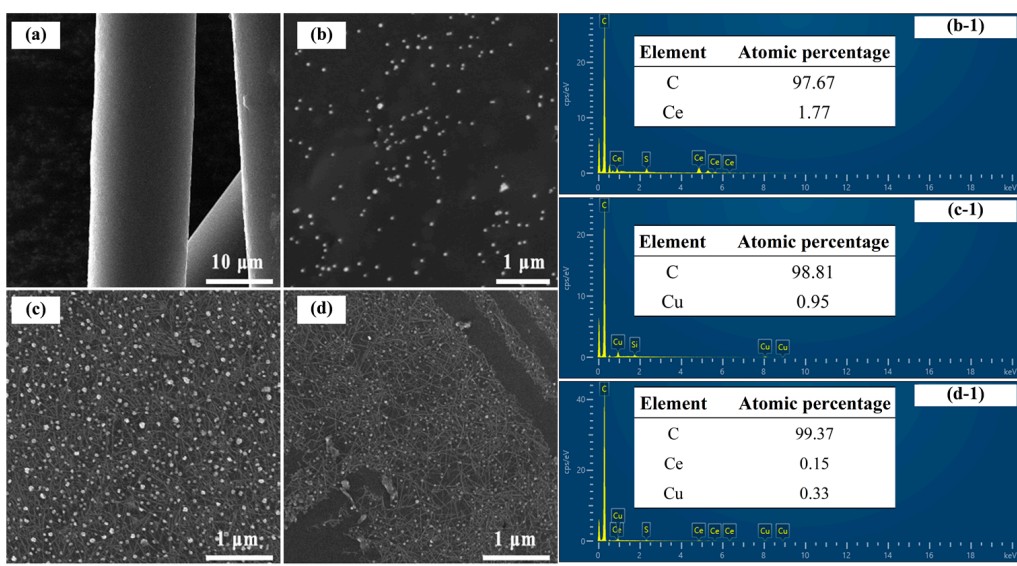

**Figure 1.** SEM–EDS images of (**a**) ACF, (**b**) N-CNF/ACF@Ce, (**c**) N-CNF/ACF@Cu and (**d**) N-CNF/ACF@Cu/Ce.

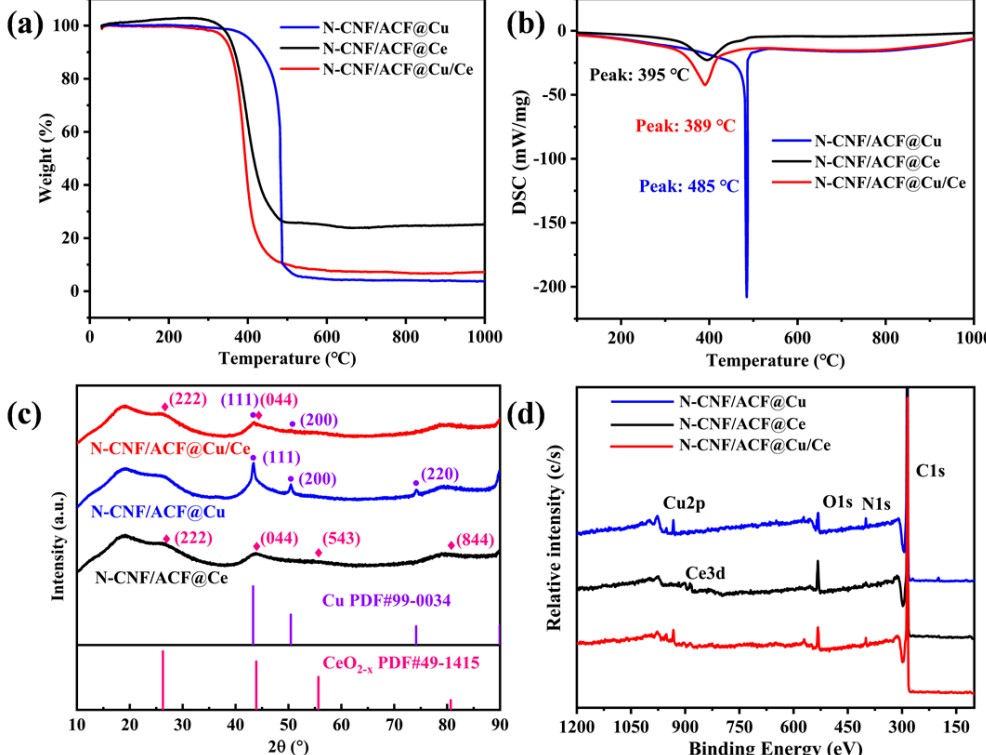

**Figure 2.** TG–DSC curves in air atmosphere (**a**,**b**), XRD patters (**c**) and XPS spectra (**d**) of N-CNF/ACF@Me.

The X-ray diffraction (XRD) analyses were carried out to analyze the phase composition of the samples. As shown in Figure 2c, the XRD patterns of the N-CNF/ACF@Cu and the N-CNF/ACF@Ce matched Cu PDF#99-0034 and $CeO_{2-x}$ PDF#49-1415, respectively, indicating that Cu and Ce were successfully transferred from the nitrate solution to the CNF/ACF surface. The characteristic peaks that were corresponding to both of the phases were observed in the N-CNF/ACF@Cu/Ce despite it having a low peak intensity, and the Cu (111) and $CeO_{2-x}$ (044) peaks were broadened due to their relatively small grain sizes.

The X-ray photoelectron spectroscopy (XPS) analyses were performed to compare the chemical changes on the fiber surface and to analyze the valence states of the metal nanoparticles, thus speculating their roles in the $NH_3$-SCR reaction. Figure 2d shows the XPS full-scan spectra of the N-CNF/ACF@Me, and Figure S1 shows the deconvolution results in the Cu2p and/or Ce3d domains, which provide information about the changes of the element composition and the electronic structure on the fiber's surface after loading the bimetallic nanoparticles. The characteristic peaks Cu2p in the N-CNF/ACF@Cu and Ce3d in the N-CNF/ACF@Ce were successfully detected, while the spectrum of the N-CNF/ACF@Cu/Ce contained both of the signals, thereby indicating that the metal nanoparticle catalysts were successfully loaded. The N-CNF/ACF@Cu had the highest N1s peak due to the growth of the nitrogen-rich CNFs, while the N-CNF/ACF@Ce had the highest O1s peak thus proving the oxygen affinity of Ce and the N1s peak, thereby suggesting the formation of nitrogen-containing functional groups. Both of the O1s and N1s peaks could be observed in the N-CNF/ACF@Cu/Ce. Table 1 shows that the ratios of $Ce^{3+}/(Ce^{3+}+Ce^{4+})$ in the N-CNF/ACF@Ce and the N-CNF/ACF@Cu/Ce were 21.48% and 22.30%, respectively, thus indicating that the synergistic effect of Cu and Ce could help produce more $Ce^{3+}$, introducing rich oxygen vacancy defects, thus providing more active sites for the oxygen chemisorption.

**Table 1.** XPS results, NO adsorption ability and $NH_3$-SCR reactivity of N-CNF/ACF@Me.

| Samples | XPS Analysis (at%) | | | NO Adsorption Ability (%) | $NH_3$-SCR Reactivity | |
| --- | --- | --- | --- | --- | --- | --- |
| | Cu2p | Ce3d | $Ce^{3+}/$ $(Ce^{3+} + Ce^{4+})$ | | NO Conversion (%) | T (°C) |
| N-CNF/ACF@Cu/Ce | 0.48 | 0.22 | 22.30 | 63.08 | 72~81 | 295~495 |
| N-CNF/ACF@Cu | 0.48 | 0.00 | – | 62.31 | 60~62 | 269~293 |
| N-CNF/ACF@Ce | 0.00 | 0.38 | 21.48 | 40.47 | 67~82 | 296~347 |

Moreover, in order to evaluate the activity retention of the metal catalysts, the $H_2$-TPR tests were conducted. As shown in Figure S2, the $H_2$ consumption peaks of the N-CNF/ACF@Ce and the N-CNF/ACF@Cu/Ce at 535 °C were attributed to the partial $Ce^{4+}$ → $Ce^{3+}$, and the peak of the N-CNF/ACF@Cu at 344 °C was attributed to the reduction of oxidized Cu. However, the peak at 344 °C could not be observed in the curve of the N-CNF/ACF@Cu/Ce, indicating the fine maintenance of $Cu^0$ on the surface.

The $N_2$ adsorption–desorption isotherms and pore size distribution curves were measured by a specific surface area tester. The Brunauer–Emmett–Teller (BET) equation was used to calculate the BET specific surface area ($S_{BET}$), and the *t*-plot method was used to calculate the micropore volume ($V_{mic}$). Figure 3a,b shows the nitrogen adsorption isotherms and the pore size distribution of the N-CNF/ACF@Me. The N-CNF/ACF@Ce had the lowest $S_{BET}$ of 599 $m^2$/g and a $V_{mic}$ of 0.26 $cm^3$/g, indicating that a serious pore blockage occurred when only Ce was loaded, but adding Cu increased the specific surface area, which proved the synergistic effect of Cu and Ce.

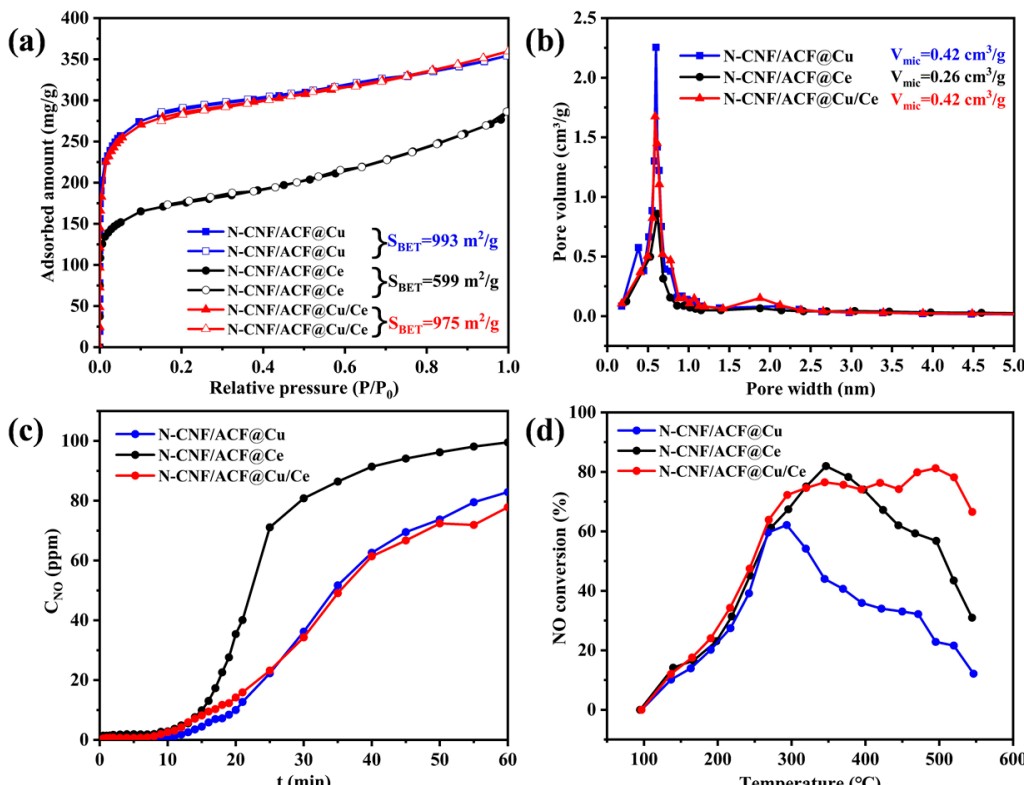

**Figure 3.** Nitrogen adsorption isotherms (**a**), pore size distributions (**b**), NO adsorption curves (**c**) and SCR reactivity curves (**d**) of N-CNF/ACF@Me.

To further investigate the adsorption–desorption behavior of $NH_3$, the $NH_3$-TPD tests of ACF and the N-CNF/ACF@Me were carried out. As reported, $NH_3$ that was desorbed at 100~200 °C, 200~500 °C and above 500 °C was adsorbed on the weak acidic sites, the moderate acidic sites and the strong acidic sites, respectively [22]. The chemical adsorption and activation of $NH_3$ on the fiber surface were very important for $NH_3$-SCR, and the amount and strength of the surface acidic sites had a significant impact on the redox reaction.

As shown in Figure S3, there were obvious $NH_3$ desorption peaks at about 900 °C in the N-CNF/ACF@Me when it was compared to the ACF. The N-CNF/ACF@Ce reached the highest desorption temperature, indicating that it had the strongest surface acidity, which might be because of the strong affinity between Ce and oxygen which generated more chemically adsorbed oxygen. The peak temperature of the N-CNF/ACF@Cu/Ce was also slightly higher than that of the N-CNF/ACF@Cu, indicating that the surface acidity of the modified fiber was enhanced, which was helpful to improve the $NO_X$ removal.

To compare the NO adsorption properties, 0.5 g N-CNF/ACF@Me was heated to 500 °C at 20 °C/min, and then, it was cleaned at a constant temperature in $N_2$ flow for 1 h. After this, the samples were cooled down to room temperature, and 100 ppm NO, $N_2$ and 10% $O_2$ were introduced into the tube at a flow rate of 100 mL/min. The concentration of NO at the outlet was monitored using an analyzer. As shown in Table 1 and Figure 3c, the CNF/ACF@Ce had the worst adsorption ability because it had a low $S_{BET}$ and a serious pore blockage occurred, thus it adsorbed only 40.47% NO in 1 h, while the other two materials had better adsorption properties due to the 3D network structure of the CNFs which provided abundant pores and sites for the NO capture. The CNF/ACF@Cu/Ce adsorbed the most amount of NO (63.08%) in 1 h, thus they had the largest NO adsorption capacity.

The $NH_3$-SCR reactivity was also evaluated. To simulate the automobile exhaust, $N_2$ was used as the equilibrium gas, and then 500 ppm NO, 500 ppm $NH_3$ and 5% $O_2$ were

mixed and then passed into the reaction device containing 0.5 g N-CNF/ACF@Me at a total flow rate of 500 mL/min, and the concentration of $NO_X$ at the outlet was detected using an analyzer. As shown in Table 1 and Figure 3d, the N-CNF/ACF@Cu/Ce, the N-CNF/ACF@Ce and the N-CNF/ACF@Cu reached the NO conversions of 72~81% at 295~495 °C, 67~82% at 296~347 °C and only 60~62% at 296~347 °C, respectively. Due to the synergistic effect of Cu/Ce, the N-CNF/ACF@Cu/Ce exhibited the most stable NO concentration at a high temperature, and it had the widest operable temperature window among all of the three materials, thus enabling N-CNF/ACF@Cu/Ce to well meet the requirements of the practical applications for NO removal, while immediate deactivation of the catalysts occurred in both the N-CNF/ACF@Ce and the N-CNF/ACF@Cu at relatively low temperatures after reaching their highest conversion. Cu and $CeO_{2-x}$ had a synergic effect on the NO reduction: Cu played a catalytic role by cooperating with the active sites of the ACFs to directly reduce NO [13,23], while $CeO_{2-x}$ acted as the oxygen source because of rich oxygen vacancies, and it increased the content of the oxygen-containing surface functional groups, thus promoting the formation of the nitrogen-containing surface functional groups during the $NH_3$ functionalization and eventually boosting the NO reduction by the $NH_3$-SCR [13,24].

Furthermore, as shown in Table S1, the N-CNF/ACF@Cu/Ce was compared with some previous works [25–29] on $NH_3$-SCR reactivity. For the comparison of the temperature windows, the temperature at which the NO conversion started to stabilize or reach 80% was chosen as the start, and the temperature at which the NO conversion started to decrease continuously was chosen as the end. Although the N-CNF/ACF@Cu/Ce achieved a relatively low level of NO conversion, the total atomic percentage of Ce and Cu in the N-CNF/ACF@Cu/Ce was only 0.48%, thereby reducing the amount of metal elements in the SCR catalysts and thus lowering the toxicity and cost. Moreover, the N-CNF/ACF@Cu/Ce had the widest operable temperature window, and it could maintain a stable NO conversion over 400 °C instead of general deactivation, thereby suggesting that is has high thermal stability in anoxic flue gases. Therefore, the catalyst that was prepared could well meet the requirements of the practical applications.

## 3. Experimental

### 3.1. Materials

The ACFs were prepared by the laboratory using the pitch as the raw material. The cerium nitrate ($Ce(NO_3)_3$), copper nitrate ($Cu(NO_3)_2$), concentrated nitric acid and anhydrous ethanol were purchased from Sinopharm Chemical Reagent Co., Ltd. (Shanghai, China). The sodium dodecyl sulfate (SDS) was purchased from Hengxing Chemical Reagent Manufacturing Co., Ltd. (Tianjin, China). All of the chemicals were used as they were received without further purification.

### 3.2. Preparation and Characterization

As shown in Figure 4, first, the pitch-based ACFs were oxidized by $HNO_3$ in a condensation reflux device and heated in an oil bath of 60 °C and magnetically stirred to obtain oxidized the ACFs (O-ACFs). Then, the O-ACFs were impregnated in the solution of copper nitrate and cerium nitrate (molar ratio 1:1, 2:0 and 0:2) and SDS in a beaker, heated in an oil bath and magnetically stirred at 60 °C for 5 h. After that, the fibers were pyrolyzed in a quartz tube at 200 °C in $N_2$ atmosphere to obtain fibers that were loaded with metal nanoparticle catalysts (O-ACF@Me). Further, the hydrogen reduction of O-ACF@Me was carried out at 300 °C in $H_2$ and Ar atmosphere to convert CuO into Cu. Then, using $C_2H_4$ as the carbon source and introducing $NH_3$ for nitrogen doping, the CVD process was carried out at 700 °C to grow carbon nanofibers (CNFs) on the surface of the ACFs, and finally, the catalytic material (N-CNF/ACF@Me) was obtained.

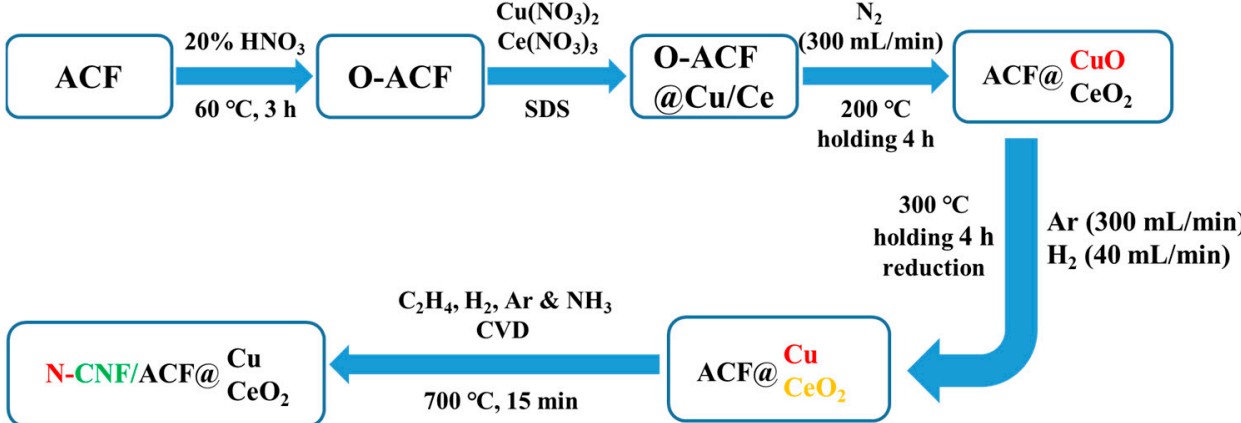

**Figure 4.** Preparation progress of the catalytic material.

The characterization of obtained samples are detailed described in the Supplementary Information.

## 4. Conclusions

By comparing the structures and properties of the N-CNF/ACF@Cu, the N-CNF/ACF@Ce and the N-CNF/ACF@Cu/Ce, it can be found that Cu played a key part as a catalyst in the growth of the CNFs during the CVD, while Ce in the samples had mixed valence states of +3 and +4, which was conducive to the redox reaction. Under the synergistic effect of Cu and Ce, the metal nanoparticles were more evenly dispersed on the fiber surface in comparison to adding Cu or Ce alone. Moreover, the synergistic effect of Cu and Ce enabled the catalysts to have a much stabler NO conversion at a much wider operable temperature window. After introducing the reducing agent $NH_3$, the N-CNF/ACF@Cu/Ce could reach a relatively stable NO conversion of 72~81% at a wide operable temperature window of 295~495 °C with high thermal stability in the $NH_3$-SCR reaction, thus indicating that the N-CNF/ACF@Cu/Ce that was prepared in this work could be applied in $NH_3$-SCR reactions at different temperatures, which provides one pathway to achieve the practical applications.

**Supplementary Materials:** The following supporting information can be downloaded at: https://www.mdpi.com/article/10.3390/catal12101192/s1, Characterization; Figure S1: XPS peak deconvolution in the Cu2p and/or Ce3d domains of (a) N-CNF/ACF@Cu, (b) N-CNF/ACF@Ce and (c,d) N-CNF/ACF@Cu/Ce; Figure S2: $H_2$-TPR curves of ACF and N-CNF/ACF@Me; Figure S3: NH3-TPD curves of ACF and N-CNF/ACF@Me; Table S1: Comparison on $NH_3$-SCR reactivity of different materials.

**Author Contributions:** S.C.: Methodology, Investigation, Writing—original draft. Z.H.: Methodology, Investigation. J.Y.: Conceptualization, Writing—review and editing, Supervision, Funding acquisition. X.C.: Writing—review and editing. J.L. (Jiahao Liu): Data curation, Formal analysis. Y.L.: Data curation, Formal analysis. J.L. (Jun Li): Writing—review and editing, Funding acquisition. All authors have read and agreed to the published version of the manuscript.

**Funding:** This research is funded by the National Natural Science Foundation for Young Scientists of China (Grant No. 51702094) and the Natural Science Foundation of Hunan Province, China (Grant Nos. 2020JJ4203 and 2019JJ50651).

**Data Availability Statement:** Data is contained within the article.

**Conflicts of Interest:** The authors declare that they have no conflict of interest.

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
