# Peer review of "Nitrogen-Doped Pitch-Based Activated Carbon Fibers with Multi-Dimensional Metal Nanoparticle Distribution for the Effective Removal of NO"

_catalysts, doi:10.3390/catal12101192_

Round 1

Reviewer 1 Report

Comment: Thanks for inviting me to review this paper titled "Nitrogen-doped Pitch-based Activated Carbon Fibers with Multidimensional Metal Nanoparticle Distribution for The Effective Removal of NO". In this paper, the authors fabricated the catalysts for NOx removal. Other researchers have already done similar works, but I found this work interesting due to wide temperature window studies with good characterizations. The presentation of this work is excellent, which will be helpful for the possible reader of the "Catalysts". Therefore, I recommend publishing this research paper in the "Catalysts", but only after a manuscript revision. Please find my comments below.

1.      Please, explain more about the advantages of this work than the previously published ones in the introduction section.

2.      Make a comparison in a table under similar conditions between this and published works, including catalysts, NOx concentration, temperature, removal efficiency, etc., and add this table in the results and discussion section.

Author Response

We would like to express our sincere thanks to you for your valuable comments and constructive suggestions. We have tried our best to revise our manuscript according to your suggestions. We resubmitted the revised manuscript which we would like to be accepted to publish in Catalysts. Revised parts in the manuscript were highlighted in YELLOW. The Response to Reviewer was provide in the attached file.

Reviewer 2 Report

In this work, pitch based activated carbon fibers (ACFs) were impregnated with copper nitrate and cerium nitrate, and then ACFs loaded with bimetallic nanoparticles (ACF@Cu/Ce) were obtained after pyrolyzation and reduction. Unfortunately, in the NH3-SCR reaction, the NO conversion of the N-CNF/ACF@Cu/Ce is not ideal(72~81%). Therefore, there are several issues that need to be addressed in order to meet the requirements of the journal.

1.     Cu and Ce are modified into the material. By what forces are they fixed?

2.     The TG results show that the skeleton of ACFs is not resistant to high temperature. What is the reason for choosing ACFs as catalyst support?

3.     Please give the mechanism of cerium species stabilizing the skeleton of ACFs.

4.     What specific advantages does ACF@Cu/Ce have over Cu-SSZ-13?

Author Response

(The authors gave the same response as above.)
